# META-LEARNING NONLINEAR DYNAMICAL SYSTEMS WITH DEEP KERNELS

## ABSTRACT

Scientific processes are often modelled by sets of differential equations. As datasets grow, individually fitting these models and quantifying their uncertainties becomes a computationally challenging task. In this paper, we focus on improving the scalability of a particular class of stochastic dynamical model, called latent force models. These offer a balance between data-driven and mechanistic inference in dynamical systems, achieved by deriving a kernel function over a low-dimensional latent force. However, exact computation of posterior kernel terms is rarely tractable, requiring approximations for complex scenarios such as nonlinear dynamics. We overcome this issue by posing the problem as meta-learning the class of latent force models corresponding to a set of differential equations. By employing a deep kernel along with a sensible function embedding, we demonstrate the ability to extrapolate from simulations to real experimental datasets. Finally, we show how our model scales compared with other approximations.

## 1 INTRODUCTION

Differential equations are mathematical models that describe the change of a function with respect to one or more variables, such as time. They play a central role in the natural and social sciences, as they provide a way to model and understand complex systems and phenomena. Differential equations are used in physics; for example, Newton's laws of motion to describe the behaviour of objects by expressing the acceleration and velocity of an object as a function of its position and other physical variables. In biology, differential equations are used to model processes such as the spread of diseases, the growth and decline of populations, and the dynamics of biochemical reactions. They provide a grounded method of making historic and future predictions of complex systems.

In a machine learning context, the modelling power of differential equations make them excellent inductive biases if incorporated properly. Latent force models (LFMs) were introduced when Lawrence et al. (2006) modelled a network of genes interacting with a common protein in the biological process of *transcriptional regulation* using a set of ordinary differential equations (ODEs). LFMs are probabilistic models that assume that the underlying dynamics of a s system can be modeled parametrically in terms of a low-dimensional latent force. Hypothetically, this enables them to handle noisy, high-dimensional, and nonlinear dynamics. However, there are computational challenges that hinder the usability of these models.

LFMs assume a joint Gaussian process prior over the latent force and observed outputs, which is determined by the model dynamics as described by the differential equations. Inferring the latent force requires computing their posterior distribution, which is analytically tractable only for a small set of problem scenarios. In the remaining cases, which tend to be nonlinear or nonstationary dynamical systems, the posterior requires some approximation. For example, Moss et al. (2021) describe an approach for approximating the posterior in a LFM using an ODE solver. While this can reliably infer latent forces for otherwise intractable problems, relying on an ODE solver is computationally costly and makes the model intractable for a range of larger scale, real-world scenarios. Deep GPs have also been investigated for solving LFMs since, as McDonald & Álvarez (2021) points out, such shallow models do not allow enough representative power for more complex systems. Crucially, existing works do not address the serious challenge of scaling up LFMs to work in the multi-task setting. It is often desired to fit many independent LFMs simultaneously; for example, in the case of genomics, we may wish to make inferences over thousands of genes or interaction subnetworks.

In this work, we propose a meta-learning approach to solve a general class of LFMs. We avoid any ODE solving step and any variational approximations by instead learning the dynamics in a deep kernel (Wilson et al., 2016). Our framework makes use of neural network representations of sets in order to produce a function embedding for each task. Specifically, we consider the Transformer (Vaswani et al., 2017) and the Fourier neural operator (Li et al., 2020). Given a task consisting of only the input mesh, for example time, and the observed functions' embedding, our model infers the associated latent force with standard Gaussian process conditioning. This makes our approach much faster than training an LFM on individual tasks. This method can model complex nonlinear dynamics and provides solutions even to multivariate problems such as partial differential equations which were previously computationally infeasible for large datasets.

## 2 PRELIMINARIES

**Gaussian processes**  Gaussian processes are stochastic processes commonly used as priors for latent functions in Bayesian machine learning models that map from inputs $\mathbf{x} \in \mathbb{R}^D$ to predictions $f(\mathbf{x}) \in \mathbb{R}$. A GP prior

$$f \sim \mathcal{GP}(m(\mathbf{x}), \kappa(\mathbf{x}, \mathbf{x}')) \tag{1}$$

is described by its mean function $m(\mathbf{x})$ and its kernel function $\kappa(\mathbf{x}, \mathbf{x}')$. The mean function is usually set to $0$ for standardised data. The kernel function may have a set of hyper-parameters $\theta$, such as the lengthscale $l$ in an RBF kernel, $k_{\text{RBF}}(\mathbf{x}, \mathbf{x}') = \exp(-\frac{l}{2}\|\mathbf{x} - \mathbf{x}'\|_2)$. Under this prior, any finite collection of points $\mathbf{f}(\mathbf{X})$ for inputs $\mathbf{X} = [\mathbf{x}_1, \mathbf{x}_2, \ldots, \mathbf{x}_N]^\top$ is normally distributed: $\mathbf{f} \sim \mathcal{N}(m(\mathbf{X}), \kappa(\mathbf{X}, \mathbf{X}))$. If the model specifies a Gaussian likelihood for observations $y$, meaning $y \sim \mathcal{N}(f, \sigma^2)$, the posterior distribution for training data $\mathbf{X}, \mathbf{y}$ is analytically tractable and given by

$$f \mid \mathbf{y} \sim \mathcal{N}\left( \kappa(\mathbf{x}, \mathbf{X})[\kappa(\mathbf{X}, \mathbf{X}) + \sigma^2 \mathbf{I}]^{-1}\mathbf{y}, \; \kappa(\mathbf{x}, \mathbf{x}') - \kappa(\mathbf{x}, \mathbf{X})[\kappa(\mathbf{X}, \mathbf{X}) + \sigma^2 \mathbf{I}]^{-1}\kappa(\mathbf{X}, \mathbf{x}') \right).$$

Moreover the marginal likelihood has a closed form expression and is given by

$$p(\mathbf{y}) = \mathcal{N}(\mathbf{y} \mid \mathbf{0}, \kappa(\mathbf{X}, \mathbf{X}) + \sigma^2 \mathbf{I}). \tag{2}$$

The availability of a closed form expression allows optimising the kernel hyper-parameters $\theta$ by maximising the marginal likelihood using gradient-based optimisation.

**Deep Kernel Learning**  Deep kernel learning as presented by Wilson et al. (2016) constitutes an attempt to combine the representation learning capabilities of deep neural networks with the non-parametric nature of Gaussian processes. A neural network is used to map an input $\mathbf{x}$ into a latent space yielding a vector $\text{NN}(\mathbf{x}) \in \mathbb{R}^D$. This representation is then fed into a base kernel $\kappa(\cdot, \cdot)$ (such as an RBF kernel) to yield the covariance between inputs $\kappa(\text{NN}(\mathbf{x}), \text{NN}(\mathbf{x}'))$.

**Latent Force Models**  LFMs incorporate explicit dynamics of differential equations in the kernel functions of Gaussian processes (GPs) in order to infer latent forcing terms (Lawrence et al., 2006; Alvarez et al., 2009). The latent force captures the underlying process and structure in the data, while being unobserved and shared amongst the outputs. The differential equation, $g$, parameterised by $\boldsymbol{\Theta}$, enforces a mechanistic relationship between inputs, $\mathbf{x} \in \mathbb{R}^T$, outputs, $\mathbf{y}(\mathbf{x}) \in \mathbb{R}^{P \times T}$, and an unobserved latent force, $\mathbf{f}(x) \in \mathbb{R}^Q$. A GP prior is assigned to the latent force, $\mathbf{f} \sim \mathcal{GP}(\mathbf{0}, \kappa(\mathbf{x}, \mathbf{x}'))$, which naturally captures biological noise and enables non-linear expressivity through kernels. Some LFM work considers multiple forces but we do not cover this due to the identifiability issues they pose. The force can be transformed by some response function $G(\mathbf{f})$,

$$\overbrace{\mathcal{D}\,\mathbf{y}(\mathbf{x})}^{\text{differential}} = \overbrace{g\big(\mathbf{y}, \mathbf{x}; \boldsymbol{\Theta}, G \circ \mathbf{f}(\mathbf{x})\big)}^{\text{differential equation}}, \tag{3}$$

where $\mathcal{D}$ is some differential operator, for example an $n^{\text{th}}$ order derivative for ODEs or partial derivatives for PDEs. An analytical expression for the covariance between outputs, $\kappa_{\mathbf{y}, \mathbf{y}'}(\mathbf{x}, \mathbf{x}')$, is possible under the necessary condition that $G$ is a linear operator. In these cases, maximum marginal likelihood yields the differential equation parameters and inference can be carried out with standard posterior GP identities (see Rasmussen & Williams (2005)). However, we often face larger datasets with a non-linear relationship to the latent force leading to computational challenges where approximations must be used.

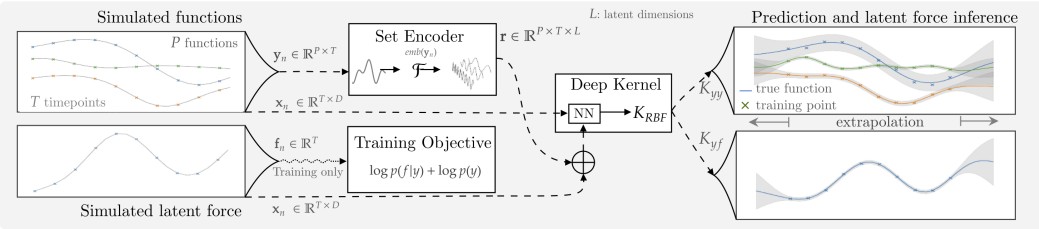

Figure 1: Schematic of DKLFM. First, a dataset of latent force tasks is created by sampling the latent force and differential equation parameters and solving the forward solution. The simulated functions are embedded by aggregating the output state of an encoder. A deep kernel is learned to represent the convolution operator of an arbitrary LFM. For training tasks, the model minimises the loss in Equation 5 with access to simulated latent force data. For test tasks, the latent force is unobserved and inferred via the cross-covariance only, as in a typical LFM scenario. The diagram shows one task; in reality, we train over batches of tasks.

## 3 DEEP KERNEL LEARNING OF LATENT FORCE MODELS

In this section, we present DKLFM (an acronym of the heading above): a novel meta-learning method for multi-task dynamical modelling. We first detail the problem setting and derive our objective function, and finally discuss any design choices in our approach.

### 3.1 MODEL FORMULATION

We assume a dataset of $N$ tasks $\{\mathbf{x}_n, \mathbf{y}_n(\mathbf{x}), \mathbf{f}_n(\mathbf{x})\}_{n=0}^N$, where $\mathbf{x}_n \in \mathbb{R}^{T \times D}$ denotes $T$ observed $D$-dimensional input points which may be temporal ($D = 1$) or spatio-temporal ($D > 1$). Our output observations, $\mathbf{y}_n \in \mathbb{R}^{P \times T}$, is the set of $P$ function outputs, and $\mathbf{f}_n \in \mathbb{R}^T$ is the latent force at the observed input points. We split the dataset into train and test tasks, where train tasks contain both the latent force data and observed outputs, while test tasks only have observed outputs.

The model setup is summarised in Figures 1 and 2. The direction in the graphical model is such that the output observations determine the latent force. While a model could have been constructed the other way around, this would not reflect reality: for real datapoints without latent force observations, we can only condition on the output functions. In order to generalise across tasks, we construct a task representation, denoted $\mathbf{r}_n = \text{emb}(\mathbf{x}_n, \mathbf{y}_n)$, where emb is an arbitrary encoder. We then learn a latent function, $h$, mapping from the input mesh, $\mathbf{x}_n$, and the task representation, $\mathbf{r}_n$, to the latent force, $\mathbf{f}_n$. Inferences are then made using GP conditioning on the output observations for arbitrary test tasks. We assign a GP prior to $h$ and use a Gaussian likelihood for the latent force, i.e.

$$h \sim \mathcal{GP}(m_h(\cdot), \kappa(\cdot, \cdot)) \qquad \text{(prior)}$$
$$f \sim \mathcal{N}(h, \sigma^2) \qquad \text{(likelihood)}$$

where $m_h$ is the mean function of the GP prior and $\kappa$ is its kernel.

Under the LFM assumptions, the distribution of $y$ is implicitly determined via the joint distribution of the latent function and the outputs. More specifically, the latent function outputs at the observed inputs $\mathbf{x}$ and the outputs $\mathbf{y}$ are jointly Gaussian distributed as

$$\mathbf{h}, \mathbf{y} \sim \mathcal{N}\left(\begin{bmatrix} \boldsymbol{\mu}_h \\ \boldsymbol{\mu}_y \end{bmatrix}, \begin{bmatrix} \mathbf{K}_{hh} & \mathbf{K}_{hy} \\ \mathbf{K}_{yh} & \mathbf{K}_{yy} \end{bmatrix}\right), \qquad \text{(joint)}$$

where the mean vectors and covariance matrices are obtained by evaluating the mean and kernel functions. As described above, the latent function $h$ operates on both the inputs $\mathbf{x}_n$ and a task representation $\mathbf{r}_n$. Its mean and kernel function thus receive a concatenation $\mathbf{z}_n = \mathbf{r}_n \oplus \mathbf{x}_n$ as input. The mean vectors $\boldsymbol{\mu}_h, \boldsymbol{\mu}_y$ are evaluations of the mean function $m_h$ of the GP prior on $h$ and a dedicated mean function $m_y$ for the outputs, as in

$$\boldsymbol{\mu}_h = m_h(\mathbf{r}_n \oplus \mathbf{x}),$$
$$\mathbf{K}_{hh} = \kappa(\mathbf{r}_n \oplus \mathbf{x}_n, \mathbf{r}_n \oplus \mathbf{x}_n). \qquad (4)$$

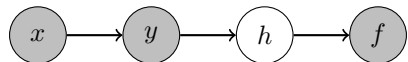

Figure 2: Graphical model

The cross-covariance $\mathbf{K}_{yh}$ and covariance $\mathbf{K}_{yy}$ are determined similarly using kernel $\kappa$, however the task representation is not present in the inputs corresponding to the index set of $\mathbf{y}_n$.

In an LFM, the kernel function is derived from a set of differential equations. In order to hold sufficient capacity in the general case where the equations cannot be solved, we define the kernel $\kappa$ to be a deep kernel (Wilson et al., 2016) with a neural network mapping the inputs to latent representation vectors before feeding them into a base kernel. In practice, this base kernel provides an additional inductive bias, for example ensuring smoothness in the latent space with an RBF kernel or periodicity with a periodic kernel. We selected constant mean functions $m_h, m_y$ with learned outputs $c_h$ and $c_y$ respectively.

The trainable parameters in our model include the task encoder weights, the deep kernel weights, and the parameters of the base kernel and mean functions. These are optimised jointly by maximising the marginal likelihood of the observed outputs and latent force for training tasks. The marginal likelihood has the closed form:

$$p(\mathbf{y}, \mathbf{f}) = \int p(\mathbf{y}, \mathbf{f}, \mathbf{h}) \, d\mathbf{h} = \int p(\mathbf{f}|\mathbf{h})p(\mathbf{h}|\mathbf{y})p(\mathbf{y}) \, d\mathbf{h}$$

$$= \mathcal{N}(\mathbf{y} \,|\, \boldsymbol{\mu}_y, \mathbf{K}_{yy}) \int \mathcal{N}(\mathbf{f} \,|\, \mathbf{h}, \sigma^2 \mathbf{I}) \mathcal{N}(\mathbf{h} \,|\, \boldsymbol{\mu}_{h|y}, \mathbf{K}_{h|y}) \, d\mathbf{h}$$

$$= \mathcal{N}(\mathbf{f} \,|\, \boldsymbol{\mu}_{h|y}, \mathbf{K}_{h|y} + \sigma^2 I) \mathcal{N}(\mathbf{y} \,|\, \boldsymbol{\mu}_y, \mathbf{K}_{yy}). \tag{5}$$

Here, $\boldsymbol{\mu}_{h|y}$ and $\mathbf{K}_{h|y}$ are defined as

$$\boldsymbol{\mu}_{h|y} = \boldsymbol{\mu}_h + \mathbf{K}_{hy}\mathbf{K}_{yy}^{-1}\mathbf{y}, \tag{6}$$

$$\mathbf{K}_{h|y} = \mathbf{K}_{hh} - \mathbf{K}_{hy}\mathbf{K}_{yy}^{-1}\mathbf{K}_{yh}. \tag{7}$$

At test time, we receive an arbitrary input $\mathbf{x}^*$. At these input locations, we define $\mathbf{h}^*$ and $\mathbf{y}^*$ as the inferred latent force and predicted outputs, respectively. We can infer the unobserved latent forces using the conditional GP defined by Equations 6 and 7. The posterior predictive distribution is used for finding the distribution over the outputs, which is the conditional GP defined by

$$\mathbf{y}^*|\mathbf{y} \sim \mathcal{N}(\mathbf{f} \,|\, \boldsymbol{\mu}_{y*|y}, \mathbf{K}_{y*|y}),$$

where the mean and covariance are found with standard GP machinery:

$$\boldsymbol{\mu}_{y*|y} = \boldsymbol{\mu}_y + \mathbf{K}_{y*y}\mathbf{K}_{yy}^{-1}\mathbf{y},$$

$$\mathbf{K}_{y*|y} = \mathbf{K}_{y*y*} - \mathbf{K}_{y*y}\mathbf{K}_{yy}^{-1}\mathbf{K}_{yy*}.$$

Note that so far we are using exact GP inference. If the input space is very large, then the matrix inversion can become computationally challenging and a variational approximation may be easily interchanged here. We will now discuss specific components of the model in more detail.

**Task Representation**  In this meta-learning problem, we simulate a dataset of tasks where the input mesh is typically the same for all instances, which would result in identical covariance matrices for our GPs. We must therefore provide the task-specific embedding, $\text{emb}(\mathbf{x}_n, \mathbf{y}_n)$, in the input space of the GP. In keeping with the LFM paradigm where latent forces are completely determined by the output dynamics, this embedding does not observe any latent force data. Instead, this representation is used by the deep kernel to learn the relationship between output functions and latent force in its latent space. As such, our embedding must contain the dynamics information usually captured by the differential equation and associated parameters. Our other requirement is input resolution invariance in order to maintain the flexibility of Gaussian processes.

To that end, in our research we explored two different encoders: a Fourier neural operator (Li et al., 2020) and a Transformer (Vaswani et al., 2017). In both cases, the embedding is a transformation

applied to the mesh dimensions ($T \times D$) of our observations, treating each of the $P$ output functions and $B$ tasks independently. We take the mean over $T$ to yield a latent vector for each output function. Our embedding function is therefore $\text{emb} : \mathbb{R}^{P \cdot B \times T \times D} \to \mathbb{R}^{P \cdot B \times L}$, where $L$ is a hyperparameter determining the size of our embedding. This is trained end-to-end with the deep kernel.

The Fourier neural operator was originally developed to solve partial differential equations (PDEs) due their mesh-invariance and roughly $\sim 1{,}000\times$ faster solving speeds. In our work, we apply a linear layer to increase the input dimensionality, $D$, followed by several *spectral convolutions*. These consist of computing the Fourier transform over the mesh size dimension and extracting the first $n$ Fourier modes, which are multiplied by learned complex weights. Finally the inverse Fourier transform takes us back into the input domain. One severe limitation is that the Fourier transform requires the input to be a regularly spaced.

In an effort to resolve this issue, we also consider a Transformer. The specific architecture used in our work is a decoder only, consisting of a linear layer followed by several layers of self-attention. The number of layers is experiment-specific and is discussed in Section 4. Sinusoidal positional encoding enabled the modelling of an irregular mesh. We found very little difference in performance between the two. Since our experiments have a non-uniform input grid, we use the Transformer.

**Resolution invariance**  Our mesh-invariance enables super-resolution inference; test cases can be at an arbitrary resolution higher than the training data, as we demonstrate in Section 4.2.

**Deep kernels**  As discussed, the kernel function in an LFM is derived from differential equations. Since we are aiming for a general approach capable of being applied to any differential equations, even PDEs, we select a simple MLP $\text{NN} : \mathbb{R}^{L_1} \to \mathbb{R}^{L_2}$. Moreover, the same network is used to transform both kernel inputs. In some experiments, particularly periodic scenarios, it helped to concatenate the input mesh after applying the neural network. This granted the periodic base kernel access to both the latent vector and input mesh. For example, based on Equation 4,

$$K_{hh} = \kappa\left(\mathbf{a}_n, \mathbf{a}_n\right) := \kappa_{\text{periodic}}(\mathbf{x}_n \oplus \text{NN}\left(\mathbf{a}_n\right), \mathbf{x}_n \oplus \text{NN}(\mathbf{a}_n)),$$

where $\mathbf{a}_n = \mathbf{r}_n \oplus \mathbf{x}_n$. Similarly for the cross-covariance, the same MLP is used. With separate networks, the model could transform the two input spaces such that the cross-covariance ignores the conditioning data entirely, leading to a poor performance on the output function.

## 4 EXPERIMENTS

In this section we investigate the performance of DKLFM on two ODE-based LFMs and one PDE-based LFM. Given that this is the first meta-learning model for latent force models, we analyse the performance on real, experimentally-derived datasets not in the synthetic training distribution.

### 4.1 NONLINEAR ORDINARY DIFFERENTIAL EQUATIONS

The first ODE model is the similar to the original application of latent force models (Lawrence et al., 2006): the biological process of transcription. In this experiment, cancer cells are subject to ionising radiation and the concentration of mRNA is measured via microarray at different timepoints. The data pertains to transcript counts for five targets of the transcription factor p53 over seven timepoints and three replicates. We also consider the paired ODE Lotka-Volterra equations, which govern predator-prey dynamics and exhibit periodic solutions.

**Latent Force Setup**  The first task models the time derivative of mRNA, $y_j(t)$, of gene $j$ related to its latent regulating transcription factor protein(s) $f_i(t)$ (Barenco et al., 2006):

$$\frac{\mathrm{d}y_j(t)}{\mathrm{d}t} = \overbrace{b_j}^{\text{basal rate}} + s_j \overbrace{G(f(t))}^{\text{response}} - \overbrace{d_j y_j(t)}^{\text{decay term}}, \tag{8}$$

where $b_j$ is the base transcription rate of gene $j$, $s_j$ is the sensitivity, or a response factor to the transcription factors, and $G$ is an optional function, for example a nonlinearity enforcing positivity or a saturation term enforcing limits on the latent force. The exact solution is only tractable when

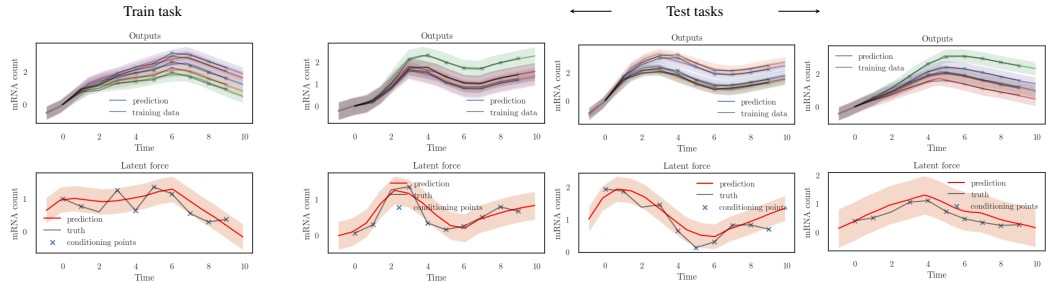

Figure 3: Training and test transcriptional regulation tasks. Notice that even for test tasks, the learned variance encapsulates most ground truth. Test tasks do not have access to latent force data.

the response function is the identity. In this case, we set $G$ to the softplus function: $G(f(t)) = \log(1 + \exp(f(t)))$. The Lotka-Volterra task is defined by the equations:

$$\frac{\mathrm{d}u(t)}{\mathrm{d}t} = \alpha u(t) - \beta u(t)v(t) \qquad \frac{\mathrm{d}v(t)}{\mathrm{d}t} = \gamma u(t)v(t) - \delta v(t), \qquad (9)$$

where $u(t)$ and $v(t)$ are prey and predator populations respectively, with growth rates $\alpha$ and $\gamma$, decay rates $\beta$ and $\delta$. The periodic kernel was used for this task in order to capture the periodic nature of the Lotka-Volterra solutions. This also improves temporal extrapolation.

We start by sampling parameters for Equation 8 from an empirical distribution of parameters learnt by running the *Alfi* (Moss et al., 2021) latent force inference package on the p53 network of genes experimentally measured by Barenco et al. (2006). This involves numerically solving the ODE. Next, the latent force is sampled from a GP prior with RBF kernel, and the ODE is solved yielding a single task. Gaussian-distributed random noise is added to the latent forces. The Lotka-Volterra dataset was simulated using a 4th-order Runge-Kutta solver. We generate 500 instances for both experiments in this fashion, and these are split into training, validation, and test tasks.

Figure 3 demonstrates that DKLFM can infer distributions over latent forces for the task of transcriptional regulation. We then apply the model trained on the simulated dataset to a real microarray dataset from Barenco et al. (2006), and show our inferred transcription factor concentration alongside the unobserved ground truth in Figure 4a. Next, we demonstrate the intra-task extrapolation in Figure 4b, where the input has been extended into the past and future. Finally, we compare our results with baseline models in Table 1.

## 4.2 PARTIAL DIFFERENTIAL EQUATIONS

In order to demonstrate the flexibility of our model, we also fit a PDE-based LFM. These are the multivariate extension of ODEs and are significantly harder to solve, with no method able to solve all classes of PDEs. Numerical solvers typically operate on a mesh and thus suffer the curse of dimensionality. Here, we demonstrate that DKLFM can be used to fit reaction diffusion equations with a very moderate dataset of 500 low-resolution tasks. The test tasks are inferred at a much higher resolution compared with at training time.

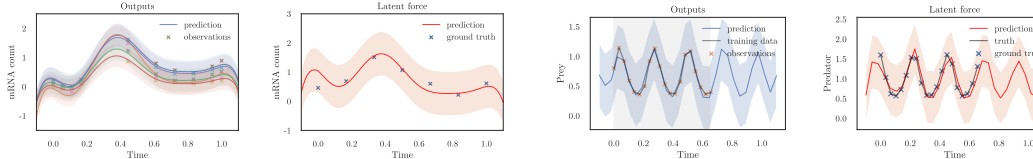

(a) DKLFM infers the protein concentration of transcription factor p53. The ground truth was published by Barenco et al. (2006). The model was trained only on simulations of Equation 8.

(b) DKLFM infers the predator-prey relationship in a Lotka-Volterra setup. The model has only been trained within the time range denoted by the grey shading, and extrapolates the periodic nature beyond.

Figure 4: DKLFM extrapolation in both tasks and input domain.

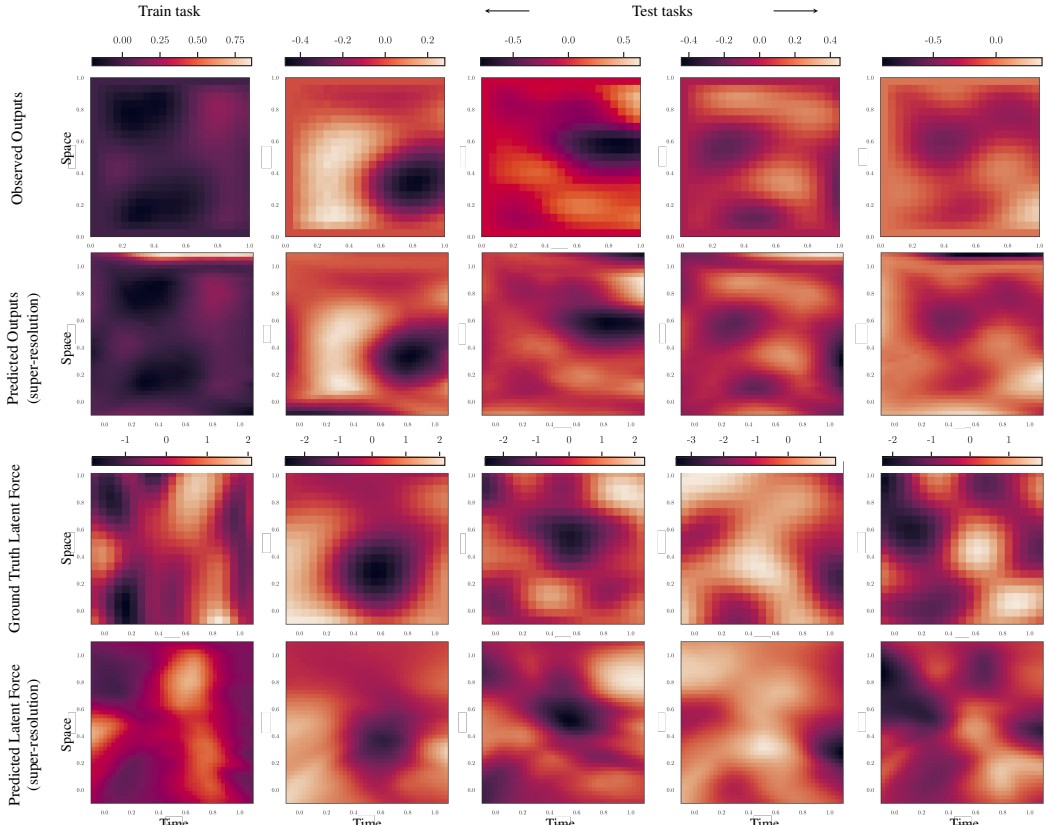

Figure 5: DKLFM trained on a synthetic reaction diffusion dataset. The first column is a training example and the next four are test cases, where the latent force is not observed. The embedding size was increased to 96 to account for the increase in dimensionality. The model was trained with a $21 \times 21$ spatiotemporal grid. At prediction time, $40 \times 40$ grid was used to illustrate the super-resolution capability. Each pair of plots vertically shares the same colorbar to enforce the same scale and accurately demonstrate inference accuracy.

**Latent Force Setup**   Reaction diffusion equations have many uses, and in this paper we look at the biological process of Drosophila embryogenesis (formation of the fruit-fly embryo). The spatiotemporal RNA expression, $y(x,t)$ of gap genes is measured using a reaction diffusion PDE from López-Lopera et al. (2019):

$$\frac{\partial y(x,t)}{\partial t} = Su(x,t) - \lambda y(x,t) + D\frac{\partial^2 y(x,t)}{\partial x^2}, \tag{10}$$

where $S$ is the production rate of the driving mRNA, $u(x,t)$ is the latent force, $\lambda$ is the decay rate and $D$ is the diffusion rate. Notice that the latent force is 2-dimensional; indeed, DKLFM can cater for any multivariate input.

In order to simulate a dataset from Equation 10, we implemented the Green's function approximation from López-Lopera et al. (2019). We then sampled parameters uniformly by empirical inspection of the gap gene dataset from Becker et al. (2013). We selected production rates in the range $[0.2, 1.0]$, decay rates in the range $[0.01, 0.4]$, diffusion rates in the range $[0.001, 0.1]$. For the latent force, we sampled the two lengthscales (corresponding to spatial and temporal dimensions) in the ranges $[0.1, 0.4]$ since both dimensions are normalised to $[0, 1]$.

A PDE solver was not required since the approximation used gives the full covariance matrix, including cross-covariances between latent force and outputs. However, the full joint covariance matrix is singular due to repeated inputs, so sampling is implemented with the eigendecomposition rather than Cholesky. We generate 500 tasks in this fashion, of which 250 are used for training.

Table 1: Comparison to baseline models for the transcriptional regulation ODE. For the *DKLFM*, we train on a dataset of 256 instances. *Alfi* and *DeepLFM* optimise each instance independently. Results are averages over 20 instances. *DKLFM* and *DeepLFM* were run on an NVIDIA GeForce RTX 4090 GPU.

| Model | Latent MSE ↓ | Output MSE ↓ | Time (s) ↓ | Mechanistic |
|---|---|---|---|---|
| Alfi | 0.117 | 0.0155 | 3.27 | Strong |
| DeepLFM | - | 0.0332 | 12.6 | Mid |
| DKLFM | **0.108** | **0.0028** | **0.0118** | Weak |

Table 2: Comparison to baseline models for the reaction diffusion PDE. The time column corresponds to the inference time per-instance.

| Model | Latent MSE ↓ | Output MSE ↓ | Time (s) ↓ | Mechanistic |
|---|---|---|---|---|
| Alfi | **0.0886** | **0.0215** | $> 10m$ | Strong |
| DeepLFM | - | 0.356 | 96.7 | Mid |
| DKLFM | 0.633 | 0.720 | **0.0523** | Weak |

We show that we can learn a general solution operator for PDE tasks in Figure 5, invariant to input resolution. In Table 2, we show how our framework compares against single-instance models. While *Alfi* obtains the most accurate result, the computational burden outweighs the benefits for any reasonably sized dataset.

## 4.3 CASE STUDY: PERFORMANCE COST

The utility of LFMs for large scientific datasets is limited by their lengthy training times. For example, in genomics, a realistic scenario is where a bioinformatician will want to train an LFM on the order of several thousand high-variance genes. This limits the use of the available approximations.

Our framework, however, solves many LFMs simultaneously rather than optimising a single instance. Therefore, an analysis of the relationship between error and training set size is key to finding the point our error rate drops to the level of solving an individual instance. If this is less than or similar to the number of instances in a typical use-case, then it is computationally preferable to generate a simulated dataset of this size rather than to train individual LFMs. For this study, we compare against *Alfi*, an accurate nonlinear LFM approximation defined in Moss et al. (2021). We chose the ODE task, since the PDE solver in *Alfi* is too computationally intensive for this comparison. In Figure 6, we confirm our hypothesis by plotting the MSE versus dataset size for our model, and horizontal lines are the mean MSE for *Alfi* over a subset of 64 tasks.

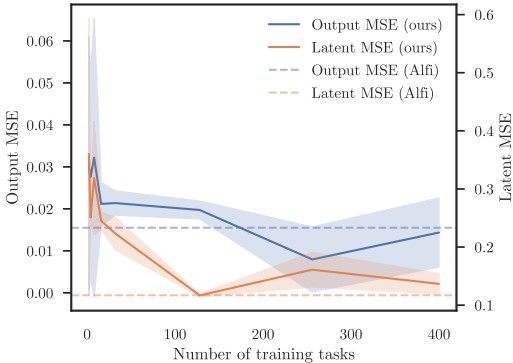

Figure 6: We plot MSEs against training dataset size to demonstrate the point at which it becomes more economical to use DKLFM rather than single-instance models. At around 200 tasks, the performance of DKLFM matches that of a single-instance model optimised by Alfi.

## 5 RELATED WORK

Differential equation-based inference in dynamical systems with Gaussian processes was introduced in Lawrence et al. (2006) and Alvarez et al. (2009). These approaches derive the kernel functions by solving the convolution integral of a base kernel with a linear operator corresponding to the ODE solution. When the dynamics becomes nonlinear, the Laplace approximation was used for the marginal likelihood. The primary issue with this technique is the manual requirement of solving the specific ODE as well as the first and second derivatives of the nonlinearity.

Ward et al. (2020) employ a state-space model for approximating the posterior of a non-linear LFM. The authors use autoregressive flows to construct a joint density of the state using variational inference, bypassing complex kernel derivations but resulting in the over-confidence prevalent in such black-box variational approaches.

Alfi (Moss et al., 2021) avoids the complex derivations of kernel functions by sampling the latent force from the GP prior and gradient matching to pre-estimate reasonable parameters. An ODE or PDE solver is then used to fine-tune the parameters with the forward solution of the equations. The use of a solver renders this approach too computationally intensive for a multi-task setting.

McDonald & Álvarez (2021) tackles non-linear and non-stationary dynamics by constructing a deep GP (Damianou & Lawrence, 2013). At each layer, an RBF kernel is convolved with the Green's function of the ODE. This deep representation enables the modeling of a wider range of tasks than a standard LFM. It is, however, not directly applicable to PDEs or to a multi-task setting.

## 6 CONCLUSION

We have introduced a novel meta-learning framework for latent force models by leveraging the expressive power of deep kernels combined with a learned task representation. Where standard LFMs require an optimisation loop to find kernel parameters, our approach only requires GP conditioning on observations at prediction time, enabling extremely fast latent force inference. Specifically, this involves inverting a $T \times T$ matrix with $O(T^2)$ computational complexity. If the input size is too large, a technique such as variational inducing points reduces the computational complexity to $O(TM^2)$ with $M$ inducing points. DKLFM is therefore an *exact inference* probabilistic model: the first of its kind for learning the solution operator for an arbitrary nonlinear LFM. We achieve this by learning a deep kernel corresponding to the differential equation by training on a simulated dataset of tasks. The embedding of each task's observations are interpreted as the task representation, containing information such as rate parameters. At test time, this representation along with an arbitrary input is used to compute the latent forces with Gaussian process conditioning on observations.

**Limitations** Speed is limited by generating the training dataset, which is easily parallelised. We envisage these models being used like large language models (Shanahan, 2022), where a user can fine-tune a pretrained DKLFM to make latent force inferences on their dataset. The original LFM is *strongly mechanistic*, deriving the covariance function from strict dynamics equations. While this may be more robust in the presence of lots of noise, it is overly rigid for real-world tasks. Our approach is *weakly mechanistic*: dynamics are not imposed, but rather parametrically learnt from the data and paired with a nonparametric GP to condition on unseen data. As with AlphaFold (Jumper et al., 2021), combining biophysical priors and data-driven approaches may be more appropriate for complex problems. The inference performance of DKLFM is reliant on learning a good cross-covariance between latent forces and observations. We hypothesise that this is why this model does not exhibit the tendency for over-confidence in predictions commonly found in related approaches. Over-confidence away from training data is also a problem for deep kernel learning, however since we learn the same deep kernel over a dataset of tasks this has proved not to be an issue.

**Further work** DKLFM explicitly treats the uncertainty in the latent forces and output functions. In an active learning context, we can query input points where the output function or latent forces have high-uncertainty. This enables experiment design, for example to determine an appropriate coarseness for a time-course experiment. Furthermore, we have currently only considered one latent force per task. Multiple forces lead to identifiability issues, where many combinations of latent forces would solve the same LFM. We therefore leave this to future research.

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
