# OpenReview forum: "Meta-Learning Nonlinear Dynamical Systems with Deep Kernels"
_ICLR.cc/2024/Conference — Submitted to ICLR 2024_

### Official Review · Reviewer_Sax4 · 2023-10-25

**Soundness:** 3 good
**Presentation:** 2 fair
**Contribution:** 3 good
**Rating:** 5
**Confidence:** 3

**Summary:**

This paper tackles the computational challenges of fitting latent force models to large datasets. These models bridge data-driven and mechanistic inferences but are computationally demanding, especially in nonlinear dynamics. The authors introduce a meta-learning approach using a deep kernel and functional embedding, enabling better extrapolation from simulations to real data. The proposed method's scalability is compared favorably against other approximation techniques.

**Strengths:**

The method is technically sound, and the problem is theoretically interesting. The use of task representation in meta-learning scheme here is interesting, and the method is novel in providing a numerical solver free approach for dynamical system solving.
The papers popularizes another point of view of dimensionality reduction method that is specific for physical models and makes use of this method to do meta-learning.

**Weaknesses:**

The paper mainly compares with [1], which is a single task approach, and is slow for using ODE/PDE solvers. The main argument made here is that the proposed approach uses inference to get rid of the numerical solvers, and uses task encoding to broaden to multi-task setting, and so when applied to many tasks at the same time (about 200 as reported in Figure 6), the proposed method will achieve both a quick inference and an accurate one. However, training the proposed model requires an expensive generation of training datasets, which is not required in [1]. I understand that the approach is trying to show task representation is useful in generalizing to fast multi-task setting, but some information on the training cost of the proposed approach compared to that of [1] would be appreciated.

Besides, the empirical results are mostly focused on very low-dimensional tasks, and are not showcasing the power of LFMs. It would be great to see some high-dimensional experiments.

[1] Jacob D Moss, Felix L Opolka, Bianca Dumitrascu, and Pietro Lio. Approximate latent force model inference. arXiv preprint arXiv:2109.11851, 2021.

**Questions:**

N/A

---

### Official Review · Reviewer_r4WP · 2023-10-29

**Soundness:** 3 good
**Presentation:** 3 good
**Contribution:** 3 good
**Rating:** 5
**Confidence:** 4

**Summary:**

Exact Gaussian processes inference has been well known for its heavy computation. This is especially an obstacle to practical use to large data set for latent force models which the latent force is modeled by GPs. This work aims to overcome this issue. The authors proposes latent force models with deep GP kernels and its fast inference algorithm. In meta-learning framework, the task representation vector is concatenated to the input vector as the input of kernel. The proposed method is tested on synthetic dynamical systems and real experimental data.

**Strengths:**

The proposed method has been shown fast and scalable to large datasets comparing to other methods, which makes more practical use of LFMs. The introduction of task representation vector is to generalize LFM to new tasks. The deep GP kernels grants the model more expressive power.

**Weaknesses:**

The model formulation needs clarification.
- Missing observation (output) $\mathbf{y}$ likelihood.
- Latent force $\mathbf{f}_n$ and $f$ (likelihood) are the same or not?
- Better use symbols consistently. The same symbols in GP preliminaries and model formulation with different meanings are confusing.

The figures captions lack description.

It is unclear how this is and should be in framework of meta-learning. What meta knowledge is learned and used, the task representation? The encoder for task representation is undermotivated.

**Questions:**

- Why are the latent force observable?
- Why do the observations not exist for test in Fig 4b?
- What do the colors represent in the figures?
- What kind of error bars are used?

---

### Official Review · Reviewer_6wso · 2023-10-31

**Soundness:** 3 good
**Presentation:** 3 good
**Contribution:** 3 good
**Rating:** 3
**Confidence:** 3

**Summary:**

This paper proposes a method for improving scalability of a particular class of stochastic dynamical model, called latent force models. To this end, the paper proposes a method, deep kernel learning of latent force models, which embeds a data instance into an instance-specific representation, which is later used to infer a hidden representation, on which a Gaussian process operates on. With the GP machinery, hidden representations of unseen input mesh can be inferred and, consequently, output can be inferred. The proposed method is tested on two classes of benchmark problems, ordinary differential equations and partial differential equations.

**Strengths:**

- The motivation is clear.

- The mathematical description of the paper is clear.

- The experimental results seem to provide the proposed model’s efficiency (or presumably the scalability) over the compared baselines.

**Weaknesses:**

- The use of terms such as “multi-task modeling” and “meta-learning” may require some careful considerations. The task in the manuscript is defined as each data instance. For example, in the dynamics modeling context, a single task here corresponds to learning a dynamics function (or inferring a parameter as in system identification) from a set of measurement, which is collected from simulating a set of ODEs with a specific parameter. This way of defining “task” may be considered as a non-standard way in “multi-task learning” and, thus, a proper definition would be needed. Relatedly, the use of the terminology “meta-learning”  also seems to require proper justification. Without these, the current method can be seen as a single framework, which is a mixture of parametric and non-parametric ML algorithms, where the parametric part of the framework is being trained on multiple input instances (which are denoted as tasks in the paper) via likelihood maximization with a regular gradient descent algorithm (without specifically learning meta-learned model parameters).

- The assumption on the availability of the latent force seems to be less practical. As shown in one of the benchmarks (the ODE benchmark), it does seem that latent force needs to be inferred from another software package (Alfi). The paper does not provide much information on how reliable this process is or if there are other alternatives for collecting latent forces (if they are not readily available). Based on this limited information, it is less convincing if the proposed method is practical. A side note is that the software, Alfi, seems to be based on the algorithm, appeared in a preprint, which has not been published; this makes it ever harder to assess the practicality of the proposed method.

- The description of the experimental section is not very clear. There seem to be two separate ODEs, mRNA and LV. But the descriptions on those two ODEs are mixed together and it’s unclear how the tasks are defined and how the actual experiments have been performed. Moreover, it is unclear if the authors provide all sufficient information on their experiments.

- An empirical side of the paper is weak. The proposed method has been tested on  relatively simple sets of benchmark problems, which does not seem to provide much insight on the effectiveness of the method or the scalability of the method. It does not seem that there is no place where the scalability of the method is discussed. Finally, the output MSE of the method on the second benchmark problem seems to be very poor.

[Update after the rebuttal] I acknowledge that I have read the authors' responses and I left my comments on the authors' responses and the revised manuscript. As in my comment, I will keep my score.

**Questions:**

- There are several questions on the weaknesses section.

- In addition to the ones already asked, here are additional questions.

  - The paper discusses the choices of the embedder. Would there be any comparisons between different choices of embedder? What are the specifications of deep kernels that are used in the experiments? Has the authors considered set encoders as well for their purpose?

  - Can the authors provide more details on the experiments? How the dataset has been created, e.g., uniform sample in the temporal domain, what integrator was used for the mRNA ODE, etc?

  - How is the latent MSE defined?

---

> ### Author Response · Authors · 2023-11-17
>
> We thank you for your detailed feedback on our paper and in advance for your time reading our response.
>
> 1. You are correct in your understanding of “task” under the meta-learning framework in our paper: it is learning the latent force and outputs of a system of parameterized equations. The task representation, or meta-parameters in meta-learning terminology, are inferred exactly via our framework. This is a key strength, since closed-form meta-losses enable gradient updates of the meta-parameters much more effectively. We also demonstrate how we perform far from the meta-train set. We believe this distinction and the specific advantages it brings to latent force inference are critical to understanding the contribution of our work, and we would further emphasize that our primary contribution is centered on latent force inference rather than meta-learning per se.
>
> 2. Regarding the assumption of the availability of latent forces. Please note that LFMs are specified as a generative process involving differentiation equations and (typically) GP-distributed latent forces. Therefore, these latent forces can be sampled, the equations solved, yielding the training dataset for a DKLFM. The latent forces are thus only accessible at training time. At inference time, like any LFM, the latent force is not available to the model.
>
> 3. We note that there may be some misunderstanding with regards to the use of Alfi, the software package we use to simulate a training dataset given the differential equations. We do not use Alfi to infer latent forces at any point. Moreover, Alfi is not based on an algorithm presented in this paper, and we don’t consider the publication status of Alfi to impact the utility of this method; indeed, if a different method was used to construct the training set, it would not impact the DKLFM.
>
> 4. Latent MSE is defined as the MSE between inferred latent forces and the simulated test dataset. This test dataset is generated from the same generative process as the training set, but with a reduced parameter range, in order to better capture extrapolative performance. We are currently working on adding NLL and ELBO to the table of results.
>
> 5. Different embeddings. In our paper we discuss two different embedders: a neural operator and an attention-based model (Transformer). They actually performed similarly for experiments with a uniform input grid. Due to this, and since some of our experiments were non-uniform, we chose the Transformer. We considered that a quantitative comparison would not have added much value. Thank you for pointing this out, and we have amended the manuscript to include this point. I’m sure a different set encoder would also be appropriate here, and if you had a specific one in mind then we would be happy to discuss.
>
> 6. While we acknowledge that the first experiment on gene interactions may be considered a simple baseline task (albeit a necessary one), we would argue that the PDE problem and periodic solution to the Lotka Volterra equations are more complex problems for the latent force model framework to solve. Indeed, LFMs are typically evaluated on simpler tasks than those discussed in this paper.
>
> 7. Moreover, with regards to scalability, we would like to highlight Section 4.3 which specifically discusses scalability of our method. Our model’s output MSE beats that of the two comparison methods for the ODE task. For the PDE task, it is important to view the metrics in combination. It takes significantly longer to obtain a better MSE with other methods. We believe these factors are key to understanding the impact of our approach for large datasets.
>
> We have made our edits to the rebuttal in red and will continue to update the manuscript as we respond to the other feedback. We look forward to an ongoing discussion with you regarding these points.

---

> > ### Comment · Reviewer_6wso · 2023-11-22
> >
> > First of all, I'd like to thank the authors for their response. After reading through the authors' response and other reviewers' comments, I came to a conclusion that I will keep the current rating.
> >
> > For some points (e.g., Point 5 in the response), I tend to agree with the authors' response. However, I feel that it might not be sufficient to meet the high bar (of ICLR and other similar venues) or to provide any scientific rigor by just saying, for example, "We found very little difference in performance between the two." as a justification of the choice made. Also, if there is very little difference, it might be advisable to make the experimental setting more challenging for FNO so that the difference can be more pronounced.
> >
> > As the concerns on other points (e.g., meta-learning interpretation, usage of alfi, scalability) are still remaining or less clearly resolved, I'd like to stick with the original assessment on the paper.

---

### Official Review · Reviewer_T2Lg · 2023-10-31

**Soundness:** 2 fair
**Presentation:** 2 fair
**Contribution:** 2 fair
**Rating:** 3
**Confidence:** 4

**Summary:**

The paper considers the problem of fitting the models of scientific processes (given by sets of differential equations) and quantifying their uncertainty. It is a challenging task, when the datasets are becoming larger. The authors focus particularly on improving the scalability of the less-known class of stochastic dynamical models - latent force models, which operates on kernel functions over a low-dimensional latent force. To overcome the challenging issue of exact computation of a posterior kernel, the paper proposes to rewrite this task into a problem of meta-learning the class of latent force models corresponding to a set of  differential equations. The main idea here is using the known Deep Kernels approach.

**Strengths:**

The paper has a few significant strengths, which I will outline below:
1. The paper considers the challenging and important problem of modeling complex scientific scenarios given a nonlinear dynamics. This is important not only from the deep learning by also other sciences perspectives.
2. The idea of improving the improving the scalability of the latent force models by incorporating the Deep Kernels approach is reasonable, especially, because the Deep Kernels and GPs enforce strong Bayesian structure.
3. The presented method seems to be faster than the other consider methods.
4. The flow of the manuscript is well-organized.

**Weaknesses:**

However, despite the strengths, the paper has a few major weaknesses. I will focus especially on the Experiments and Related Works sections.

**Major weaknesses:**

1. The authors place their work within the Meta-Learning field, but they do not compare the proposed model with any other Meta-Learning method. Moreover, the lack of even mentioning the Meta-Learning approaches within the Related Works section. Unfortunately, it is an important issue, because the Meta-Learning field is known from considering many GP-based approaches. To mention just a few: [1], [2], [3], [4], and [5]. I strongly suggest to compare the proposed method with other Meta-Learning approaches. It could also be a good source of new ideas for the future work.
2. If I understand correctly, the authors for each experiments used the same kernel - RBF. However, in the Deep Kernel setting, we can utilize any kernel function we would like - even the scalar product in an embedding space is able to imitate many other kernels. From my perspective, the lack of comparison across various kernels (e.g., family of Matern kernels, spectral or cosine kernel) is another important issue. Even if the RBF kernel would be the best for all of the presented experiments, I would like to see any justification or ablation study proving that.
3. The DKLFM method is compared with only two other methods: DeepLFM and Alfi, from which one is providing the exact solutions. However, taking into consideration the Results in Table 2, we can see that even the DeepLFM is significantly better than DKLFM. I would like to see incorporated DeepLFM into similar comparison as presented in Figure 6.

**Minor weaknesses:**

1. I am not particularly sure if the MSE is the best metric to compare between the solutions. The interesting will be to see how it looks like in different measures, like NLL/ELBO, since it should be available in the GP setting.
2. I am not certain if I understand correctly the comparison between Alfi, DeepLFM and DKLFM. What is the size of datasets on which each result in those tables are computed? If I understand it looks like this: the results for Alfi and DeepLFM are from the set of 20 examples and the results for DKLFM is from the set of 256 examples? Moreover, regarding the computation times if they are averaged across all examples? Please if you could elaborate more on this? If the results for 2 methods are from the smaller set of examples than for the last method, it will not be a fair comparison.


**References:**

[1] Snell, J., & Zemel, R. (2020, October). Bayesian Few-Shot Classification with One-vs-Each Pólya-Gamma Augmented Gaussian Processes. In International Conference on Learning Representations, 2020.

[2] Patacchiola, M., Turner, J., Crowley, E. J., O'Boyle, M., & Storkey, A. J. (2020). Bayesian meta-learning for the few-shot setting via deep kernels. Advances in Neural Information Processing Systems, 33, 16108-16118.

[3] Wang, Z., Miao, Z., Zhen, X., & Qiu, Q. (2021). Learning to learn dense gaussian processes for few-shot learning. Advances in Neural Information Processing Systems, 34, 13230-13241.

[4] Sendera, M., Tabor, J., Nowak, A., Bedychaj, A., Patacchiola, M., Trzcinski, T., ... & Zieba, M. (2021). Non-gaussian gaussian processes for few-shot regression. Advances in Neural Information Processing Systems, 34, 10285-10298.

[5] Chen, W., Tripp, A., & Hernández-Lobato, J. M. (2022, September). Meta-learning adaptive deep kernel gaussian processes for molecular property prediction. In The Eleventh International Conference on Learning Representations.

**Questions:**

I would like to see especially the following experiments and improvements:
1. Comparing with other Meta-Learning methods and placing the DKLFM in a right place in this field (adding needed related works).
2. Please if you could follow the experiment being an ablation study across different kernels for the Deep GPs?
3. Please, compare the DeepLFM with DKLFM in a similar regime to the one presented in Figure 6.
4. Elaborating more on the comparison between Alfi, DeepLFM and DKLFM since the experiment section is not crystal clear.


**Questions:**
1. It will be really helpful if you will be able to add the comparison on other metric like NLL or ELBO.
2. I have a question if in more complex examples the incorporation of the GPs is a good idea? I mean that the GPs are really powerful method, but it has the strong assumptions regarding the Normal distribution, which might reduce the model flexibility in a more complex (e.g., non-gaussian) distributions. It could be a good idea to incorporate the flow-based approaches which are able to map one distribution into another. Or just use the neural ODE. Out of the topic is the question what Alfi use as a particular ODE solver? Maybe it will be able to rewrite this setting into neural ODE.

---

> ### Author Response · Authors · 2023-11-16
>
> We appreciate the time and effort you have invested in evaluating our paper. Thank you for the insightful comments and suggestions.
>
> 1. Regarding Q1: we acknowledge your observation regarding the positioning of DKLFM in the context of meta-learning, and we would also like to thank you for sharing the meta-learning references. In response, we would like to clarify that our primary contribution is centered on latent force inference rather than meta-learning per se. Using meta-learning terminology,  the meta-parameters here are the task representations. The exact inference over these parameters is a key strength, since closed-form meta-losses enable gradient updates of the meta-parameters much more effectively. We also demonstrate how we perform far from the meta-train set. We believe this distinction and the specific advantages it brings to latent force inference are critical to understanding the contribution of our work.
>
> 2. Regarding Q2, and the issue of using only the RBF kernel. We recognise that this was not made clear in the manuscript. To clarify, our paper does describe scenarios of periodic nature. Specifically, at the top of page 4 and in the middle of page 5, we outline where one might wish to use a periodic kernel. Furthermore, the practical application of the periodic kernel is demonstrated in one of our key experiments, as illustrated in Figure 4b. We apologize that this was not made clearer in the manuscript, and we have revised the manuscript to clarify this.
> 3. Regarding Q3 and Q4: in our paper, we emphasize that DKLFM is set apart from DeepLFM in two fundamental aspects. Firstly, DKLFM is multi-task, whereas deepLFM only operates on a single instance. Secondly, DKLFM infers the latent forces in addition to the output functions. In contrast, DeepLFM only yields output functions. Without significant modification to the DeepLFM codebase, we would not be able to get DeepLFM to perform favorably in terms of time taken for a batch of instances, since it is currently set up for single-instance inference. The exact solutions deriving the training data are from Alfi.
> 4. While the output MSE for DeepLFM is superior to DKLFM, our evaluation is not focussed on that singular metric. For use cases we have discussed in the paper, we believe that inference time is just as important, and sometimes is a dealbreaker for usability.
> 5. Question about dataset sizes and the results tables: Table 1’s caption is worded poorly. The MSE is averaged over the same 20 test instances for each model, including the DKLFM. Computation time is also the average across the 20 test instances. We have also clarified this caption in the rebuttal revision.
>
> With regards to your second set of questions:
>
> 1. We are currently working on adding NLL and ELBO to the table of results.
> 2. You correctly point out the inductive bias that GPs bring to a model. In our view, the utility and drawbacks of this are very much application-dependent. Incorporating differential equations is a similarly strong bias, which could be replaced with a neural ODE. However, this takes us back to the discussion around what level of dynamical bias is needed in a model. In some applications, such as biophysical models, where parameters have real meaning, this bias is useful. In addition, where we know that our latent forces should be smooth, then a GP is an appropriate method to use.
>
> We have made our edits to the rebuttal in red and will continue to update the manuscript as we respond to the other feedback. We look forward to an ongoing discussion with you regarding these points.

---

> > ### Comment · Reviewer_T2Lg · 2023-11-20
> >
> > I want to thank the Authors for their response and revision of the paper. I went through all the added changes in the manuscript and the response. Unfortunately, my concerns and indicated weaknesses weren’t properly addressed in the revised version of the paper.
> >
> >
> > In the following paragraphs, I will go through the points from the authors response to my review.
> >
> >
> > 1. I understand that the primary contribution is centered on latent force inference. However, I have two very strong concerns regarding this point. Firstly, the authors didn’t mention any of the meta-learning approaches that utilize the concept of deep kernels. Some of them, like [5] from my review, are also incorporating this framework to deal with the real-world problems coming from science (i.e., molecular property prediction). I don’t expect comparison against those methods but at least placing your work properly. Otherwise, a reader might have an impression that the authors proposed novel methodology in the context of meta-learning (which is not true) even if the authors claim that their contribution is centered on latent force inference. My second concern is about using the Meta-Learning terminology (and strictly connects to the first concern). If the authors are trying to centered their work on latent force inference then why they use the term "Meta-Learning" at the very beginning of the title and then don’t compare or even place their work in the context of Meta-Learning approaches. One way or another. Maybe the better option will be to remove all the connections to the Meta-Learning, since it’s not the primary field of this work? Otherwise, a potential reader might be really concerned regarding mismatched between the paper’s title and its content and might consider the "Meta-Learning" in the title as just a catchy word.
> >
> >
> > 2. Thank you for including the information about the periodicity of the problem. I supposed that is why the authors use the RBF kernel only. However, I would like to see experiments with other kernels also (like the mentioned Matern family). It is know that almost every kernel might be given with embedding of x and x’ in some Hilbert space and by taking the scalar product there. Maybe it is also worth to be tried? The main reason why I’m asking for checking another kernels is the clear fact (e.g., in Table 2) that RBF might not be the best choice for the reaction diffusion PDE experiment, being much worse than any other baseline.
> >
> >
> > 3. Thank you for your explanations.
> >
> >
> > 4. I cannot agree with the authors, at least based on the proposed experiments. I will once again come back to experiment from Table 2, where the proposed method is much faster than Alfi but gets much worse MSE. I do not agree with the authors claim, because they didn’t include any dataset or experiment feasible for the latent force inference, where the computation time will be really unreasonable. In the experiment from the Table 2, we have only 256 instances dataset and such options: 1) Alfi - don’t need to be trained, assuming 10 minutes per instance, we have the results within 2 days, results are really close to the GT data; 2) DKLFM - has to be trained, really fast, but much worse results. I would like to see any setting where it will not be feasible to have comparable or better results in reasonable amount of time.
> >
> >
> > 5. Thank you for your explanations.
> >
> >
> > I also really appreciate the discussion with my second list of questions.
> >
> >
> > Overall, I thank the Authors for the response and including the revised version of the manuscript. I agree that the proposed method is interesting. However, I think that centering method within the Meta-Learning is a mistake. Moreover, my concerns about the comparison to baselines or trying different kernels for the GP weren’t addressed properly. Unfortunately, in this situation, I just cannot raise my score.

---

### Meta-Review · Area_Chair_YX4W · 2023-12-05

**Metareview:**

This paper considers applying latent force models to scientific data simulated from ODEs and PDEs. To address the challenge of computing the exact posterior for larger and higher dimensional datasets, the paper proposes the use of deep kernels combined with task embeddings, whether training is done in a meta-learning set-up.

While the proposed approach does achieve substantial improvements over previous versions of latent force models, the main concern raised by the reviewers is regarding the scope of the paper. From modern meta-learning point of view this paper lacks comparisons and discussions with many existing meta-learning methods. Also this paper lacks ablation studies to justify the individual choices of the components in the proposed method.

**Justification For Why Not Higher Score:**

Contributions not well justified, experiments not solid enough.

**Justification For Why Not Lower Score:**

N/A

---

### Decision · Program_Chairs · 2024-01-16

Reject